# Mechanically Tough and Conductive Hydrogels Based on Gelatin and Z–Gln–Gly Generated by Microbial Transglutaminase

**DOI:** 10.3390/polym16070999

**Published:** 2024-04-05

**Authors:** Zhiwei Chen, Ruxin Zhang, Shouwei Zhao, Bing Li, Shuo Wang, Wenhui Lu, Deyi Zhu

**Affiliations:** State Key Laboratory of Biobased Material and Green Papermaking, Faculty of Light Industry, Qilu University of Technology (Shandong Academy of Sciences), Jinan 250353, China; w36000@163.com (Z.C.); aurorachang714@163.com (R.Z.); zsw18834407603@163.com (S.Z.); lbing20000309@163.com (B.L.); 10431230390@stu.qlu.edu.cn (S.W.); whlu@qlu.edu.cn (W.L.)

**Keywords:** gelatin-based conductive hydrogels, mechanically tough, π–π stacking interactions, enzymatic crosslinking, strain sensor

## Abstract

Gelatin-based hydrogels with excellent mechanical properties and conductivities are desirable, but their fabrication is challenging. In this work, an innovative approach for the preparation of gelatin-based conductive hydrogels is presented that improves the mechanical and conductive properties of hydrogels by integrating Z–Gln–Gly into gelatin polymers via enzymatic crosslinking. In these hydrogels (Gel–TG–ZQG), dynamic π–π stacking interactions are created by the introduction of carbobenzoxy groups, which can increase the elasticity and toughness of the hydrogel and improve the conductivity sensitivity by forming effective electronic pathways. Moreover, the mechanical properties and conductivity of the obtained hydrogel can be controlled by tuning the molar ratio of Z–Gln–Gly to the primary amino groups in gelatin. The hydrogel with the optimal mechanical properties (Gel–TG–ZQG (0.25)) exhibits a high storage modulus, compressive strength, tensile strength, and elongation at break of 7.8 MPa at 10 °C, 0.15 MPa at 80% strain, 0.343 MPa, and 218.30%, respectively. The obtained Gel–TG–ZQG (0.25) strain sensor exhibits a short response/recovery time (260.37 ms/130.02 ms) and high sensitivity (0.138 kPa^−1^) in small pressure ranges (0–2.3 kPa). The Gel–TG–ZQG (0.25) hydrogel-based sensors can detect full-range human activities, such as swallowing, fist clenching, knee bending and finger pressing, with high sensitivity and stability, yielding highly reproducible and repeatable sensor responses. Additionally, the Gel–TG–ZQG hydrogels are noncytotoxic. All the results demonstrate that the Gel–TG–ZQG hydrogel has potential as a biosensor for wearable devices and health-monitoring systems.

## 1. Introduction

Hydrogels are a class of three-dimensional network materials formed from hydrophilic polymers by physical, chemical or biological crosslinking strategies. These materials have been widely used in the fields of biomedical research and tissue engineering and as cell carriers due to their reversible water absorption capacity, stability and large specific surface area, which resemble the characteristics of biological tissues [1,2,3,4]. In addition, the porous and crosslinked three-dimensional network structure of hydrogels can be used as an ionic conductor to realize long-distance transmission of high-frequency electrical signals [5,6], laying the foundation for the preparation of conductive hydrogels.

In addition to the inherent properties of traditional hydrogels, conductive hydrogels have good electrochemical properties; thus, they have attracted substantial attention [7,8]. Conductive hydrogels have a wide range of applications in biomedical engineering such as neural prostheses, controlled release systems, neural probes, electronic skins and biosensors, because of their ability to undergo reversible changes in structure, properties, and functionality in response to electric field stimuli [9]. Researchers have achieved high tensile strength, self-healing, and high sensitivity through specific synthesis strategies [10], including self-assembly [11,12] and double network [13,14] techniques. Despite these advancements, the biocompatibility and cytotoxicity of hydrogels remain key considerations in their development for biomedical applications. Therefore, the application of non-toxic and environmentally friendly approaches for the synthesis of conductive hydrogels has attracted considerable interest. In this regard, the use of natural polymers, such as polysaccharide-based polymers (i.e., cellulose, chitosan, hyaluronic acid and alginate) and protein-based polymers (i.e., collagen, gelatin and silk), is attractive because of their excellent biocompatibility, biodegradability, and non-toxic properties. Among these materials, gelatin is considered to be the biopolymer with the highest potential for biomaterial construction.

Gelatin is a polypeptide obtained by disrupting the triple helix structure of natural collagen and is the main protein in the extracellular matrix. Gelatin shares similar structures with collagen and is characterized by non-toxicity, biocompatibility and the presence of cell binding motifs (e.g., the arginine–glycine–asparagine sequence). In addition, the success of using gelatin as a biomaterial is largely due to its inherently low antigenicity and immunogenicity. Importantly, gelatin is an amphoteric electrolyte with various ionizable groups on its side chains, which serve as potential sites for a variety of conjugation opportunities. The resulting gelatin-based hydrogel acts as a semisolid electrolytic medium for delivering ions. Conductive hydrogels based on gelatin have been studied extensively, with a focus on their preparation and applicability. Han [15] developed a self-healing, hemostatic, and electrically conductive gelatin-based adhesive hydrogel, demonstrating its potential in tissue adhesives, wound dressings, and wearable devices. Wu [16] fabricated a conductive gelatin methacrylate-polyaniline hydrogel that maintained biomimetic and biocompatible properties and could be printed in complex geometries. These scholars demonstrated the superior electrical properties and biocompatibility of these materials, allowing for their potential use in tissue engineering and bioelectronic applications.

Although several favorable advances have been made in the preparation and application of gelatin-based conductive hydrogels, there are still several limitations. These restrictions include a narrow operating temperature range, low electrical conductivity, and poor mechanical strength (e.g., low elastic modulus and brittle failure) [17,18,19,20]. A common method of improving the mechanical properties of gelatin-based hydrogels is to introduce covalent crosslinks between the gelatin chains. In most cases, crosslinking does not increase the brittleness of the hydrogels, and there is a risk of cytotoxicity due to the use of chemical crosslinking agents. Research has shown that by incorporating reversible noncovalent interactions into crosslinked hydrogels as sacrificial bonds (including hydrogen bonding, hydrophobic interactions, metal ionic interactions and π–π stacking interactions), the mechanical strength and toughness of hydrogels can be significantly improved [15,21]. Qie [22] prepared a conductive hydrogel by introducing tannic acid into a gelatin hydrogel, and the hydrogel exhibited high toughness and fatigue resistance due to the π–π stacking interactions between tannic acid and the hydrogel matrix. Furthermore, π–π stacking interactions are known to provide long-range electron transport [23] and have been used to enhance polymer conductivity [22,24].

In this work, a simple strategy for the preparation of conductive hydrogels was developed to simultaneously improve the mechanical and conductive properties of gelatin-based hydrogels. As shown in Figure 1, the prepared hydrogel was formed from gelatin and carbobenzoxy-l-glutaminylglycine (Z–Gln–Gly) in water–DMSO mixtures using microbial transglutaminase (MTG) as a crosslinking agent. MTG is a versatile enzyme that can catalyze the formation of stable isopeptide bonds between glutamine residues and primary amines in gelatin chains, and it has been widely used to prepare gelatin-based hydrogels [25,26]. Z–Gln–Gly is an efficient acyl donor substrate of mTG, and it can be covalently attached to the ε-amino group of lysine in gelatin chains. Then, carbobenzoxy groups are introduced to create π–π stacking interactions [27], which, in coordination with covalent crosslinking between the gelatin chains, promote the mechanical properties of the hydrogel. Moreover, the π–π interactions between polymer chains are beneficial for electrical conductivity. In addition, the newly introduced carboxyl groups of Z–Gln–Gly can act as ion donors to improve the conductivity of the hydrogel. The structure of the developed gelatin-based hydrogel (Gel–TG–ZQG) is characterized, and the mechanical properties are studied. Furthermore, the swelling behavior and cytotoxicity of the resulting materials are investigated. Subsequently, the electrical conductivity and strain sensitivity are investigated to assess the feasibility of the fabricated hydrogel.

## 2. Materials and Methods

### 2.1. Materials

Gelatin (type A; from porcine skin, 300 g bloom) was purchased from Sigma-Aldrich. Carbobenzoxy-l-glutaminylglycine (Z–Gln–Gly) and fetal bovine serum (FBS) were both purchased from Shanghai Macklin Biochemical Co., Ltd., Shanghai, China. Transglutaminase (mTG, 200 U/g) was obtained from Shanghai Yuanye Bio-Technology Co., Ltd., Shanghai, China. Human cervical cancer cell line HeLa cells and DMEM were obtained from Wuhan Prosser Life Technology Co., Ltd., Wuhan, China. All other reagents were local products of analytical grade and were used as received without further purification.

### 2.2. Preparation of Gelatin-Based Hydrogels

The gelatin-based conductive hydrogel was prepared according to the following procedure. First, 0.79 g of gelatin was added to 8 mL of deionized water and allowed to swell for 2 h at room temperature. After heating in a constant-temperature water bath to 45 °C, the gelatin was completely dissolved. The required amount of Z–Gln–Gly was dissolved in 1 mL of DMSO. The two solutions were subsequently mixed together, and the mixture was adjusted to pH 7 with 0.1 M NaOH. The crosslinking reaction was performed at 50 °C for 60 min by adding 1 mL of a preprepared mTG solution containing 7 U of mTG, followed by a heating step of 95 °C for 5 min to inactive the enzyme. The primary amine content of gelatin was determined using the o-phthalaldehyde (OPA) method and the result was 630.42 μmol per g gelatin. According to the molar ratio of Z–Gln–Gly to the primary amines of gelatin (as shown in Table 1), the as-prepared hydrogels were correspondingly named Gel−TG, Gel–TG–ZQG (0.05), Gel–TG–ZQG (0.125), Gel–TG–ZQG (0.25), Gel–TG–ZQG (0.5), Gel–TG–ZQG (0.75), and Gel–TG–ZQG (1). Here, for simplicity, Z–Gln–Gly was defined as ZQG in gel nomenclature.

### 2.3. Characterization

The multispeckle diffusing wave spectroscopy (MS–DWS) technique was used for microrheological characterization during hydrogel formation. The MS–DWS technique is based on real-time tracking of changes in particle motion under the influence of thermal energy (via dynamic light scattering) [28]. This optical method was applied to determine the root mean square displacement (nm^2^) of particles in a medium. The gel formation and network structure of a sample could be estimated over time from the root mean square displacement. In this experiment, 20 mL hydrogel reaction mixtures were poured into glass tubes, which were subsequently placed in the Rheolaser^®^ Master (Formulaction, Toulouse, France) equilibrated at 50 °C. Measurements were taken every 1 h. Rheotest software (RheoSoft Master_v1.4.0.1-GG-beta) was used to collect and analyze the original data to obtain various rheological parameters, including the mean square displacement (MSD; nm^2^), elasticity index (EI, nm^−2^) and macroscopic viscosity index (MVI; nm^−2^).

The thermal degradation of the hydrogels was evaluated via thermogravimetric analysis (Mettler Toledo Instrument, Zurich, Switzerland) at a scanning rate of 10 °C/min and a nitrogen flow rate of 60 mL/min. Approximately 10 mg of sample was heated from room temperature to 800 °C in a small alumina crucible. The hydrogel samples were frozen at −80 °C for 4 h and freeze-dried (−50 °C, 24 h), followed by quenching in liquid nitrogen. The surface morphologies of the hydrogels were subsequently observed by scanning electron microscopy (Phenom Pure+, Phenom World, Eindhoven, The Netherlands). The pore size distribution was then determined according to the self-contained PoroMetric application.

### 2.4. Dynamic Rheology

The rheological behavior of the gelatin-based hydrogels was investigated using a rotating rheometer (ARES/G2, TA, Waters Corporation, New Castle County, USA) in dynamic mode, on which a parallel plate with a diameter of 20 mm and a gap size of 1.0 mm was mounted. Temperature sweeps were performed by heating from 5 °C to 40 °C at a rate of 0.5 °C per min at a fixed frequency of 1 Hz with a constant strain of 0.5%. Dynamic frequency scanning was carried out at 15 °C with a frequency ranging from 0.1 to 100 Hz and a constant strain of 0.5%. The storage modulus (G′) at different frequencies was recorded, and all assays were performed within the linear viscoelastic region. The hydrogel samples were prepared in a PTFE mold with a diameter of 20 mm and a height of 2 mm, and a small amount of silicone oil was applied to the surfaces of the hydrogels prior to the test to prevent water evaporation.

### 2.5. Mechanical Property Tests

The compression performance of the hydrogels was tested using a texture analyzer (TA; XT plus, Stable Micro System, London, UK) equipped with a cylindrical probe (P/0.5). The hydrogel samples were refrigerated at 4 °C for 18 h. The deformation level was 80% of the original height of the sample at a speed of 5 mm/min with a load cell of 10 kg. Each group of samples was tested three times, and the average value was recorded.

The tensile properties (tensile strength and elongation at break) of the hydrogels were measured using a material-testing machine in tensile mode (AI-3000, Gotech, Taiwan, China). Each hydrogel was cut into a dumbbell shape (4 mm × 50 mm) and tested at a speed of 50 mm/min at 25 °C. The sample groups were measured three times in parallel, and the average tensile property values were taken.

### 2.6. Swell Behavior Test

The Gel–TG–ZQG hydrogels were first prepared in a cylindrical mold with a height of 2 mm and a diameter of 20 mm and then lyophilized. The lyophilized samples were soaked in 100 mL of PBS buffer solution (0.05 M, pH 7.4) for 24 h. The swollen hydrogel samples were weighed after removal of surface water using filter paper. In this research, each group of data was measured three times, and the results were averaged. The results were calculated using the following equation:(1)$$ESR\left(\%\right)=\frac{W_{s}-W_{d}}{W_{d}}\times 100\%$$
where *W_s_* and *W_d_* are the mass of the hydrogel after reaching swelling equilibrium and the quality of the dry gel before swelling, respectively.

### 2.7. Cytotoxicity Assays

HeLa cells were used to evaluate the cytotoxicity levels of the prepared gelatin-based hydrogels through the MTT method. The lyophilized hydrogel samples were ground into fine powder with a freezing grinder (Retsch Cryomill, Haan, Germany) and sterilized with UV light for 2 h. Subsequently, 30 mg of powder was suspended in 5 mL of PBS (pH 7.4) for 12 h. HeLa cells were seeded in 96-well plates at a density of 8 × 10^3^ cells per well. Then, 20 μL of leaching solution was added to each well, and the cells were cultured in DMEM supplemented with 10% fetal bovine serum (FBS) at 37 °C in a humidified incubator. After different incubation times (24 h and 48 h), the optical density (OD) was determined by monitoring the absorbance of the broth at 570 nm using a microplate reader (SpectraMax M5, Molecular Devices, San Jose, CA, USA). Cell viability was calculated by normalizing the OD of each sample to that of the control without any addition.

### 2.8. Conductivity and Strain-Sensing Measurements

The conductivity of the hydrogel was measured using the four-point probe technique (RTS-8, Guangzhou Four Probe Electronic Technology Co., Ltd., Guangzhou, China) according to Cheng et al. [29]. The test temperature was set to room temperature.

The sensing property of strain resistance of each Gel–TG–ZQG hydrogel was monitored using a Keithley 2400 source meter to record the real-time resistance signal. The hydrogel samples (diameter: 20.0 mm, height: 2 mm) were assembled into a strain sensor with two layers of conductive copper sheets tightly fixed with copper wires at both ends of the hydrogel sample. To track human actions, the hydrogel devices were attached to the throat, fist, knee joints, and fingers using commercial insulating tape. The relative rate of change in resistance was calculated using the following formula:(2)$$\frac{∆R}{R}\left(\%\right)=\frac{R-R_{0}}{R_{0}}\times 100\%$$
where *R*_0_ and *R* are the original resistance and the resistance under pressure, respectively.

The strain sensitivity of the hydrogel sensor was quantified as the ratio of the relative resistance change (Δ*R*/*R*_0_) to the pressure (P), calculated from Equation (3):(3)$$GF=\frac{∆R}{R_{0}P}$$

## 3. Results

### 3.1. Preparation of Gel–TG–ZQG

Enzymatically crosslinked gelatin-based hydrogels were synthesized by introducing Z–Gln–Gly into the gelatin matrix (Figure 1). By performing microrheological measurements based on multispeckle diffuse wave spectroscopy (MS–DWS), the Brownian motion of scattered particles in the reaction mixtures was monitored during hydrogel formation, thus providing some rheological information on viscoelasticity. Figure 2 shows the MSD signal during hydrogel formation at 50 °C. For Gel−TG (Figure 2a), during the initial stage of the reaction (blue lines), the MSD curves were approximately linear, indicating that the particles could freely move in the reaction mixture. As the reaction progressed, the MSD curves gradually shifted downward in a nonlinear trend (green lines), and a plateau zone began to emerge, indicating that the particles were gradually encapsulated in the crosslinked network of gelatin chains (as shown in Figure 2h), resulting in a decrease in Brownian motion [30,31]. With further extension of the reaction time, the height of the plateau region of the MSD curves gradually decreased, indicating that further formation of the gel network structure occurred via enzymatic crosslinking interactions. Finally, the MSD curve became a horizontal line with a slope close to 0, indicating that the tracer particles were trapped in the hydrogel network.

When Z–Gln–Gly was introduced into the hydrogel system, the distribution of the MSD curves changed significantly (Figure 2b–d). Evidently, the addition of more Z–Gln–Gly resulted in the tracer particles being in free motion for a longer period of time, thereby increasing the time required for the MSD curve to become a horizontal line. This finding suggested that in the initial stage of the enzymatic reaction, a grafting reaction of Z–Gln–Gly on the gelatin peptide chains and relatively few crosslinking reactions between the gelatin chains occurred. A possible explanation could be that Z–Gln–Gly is a more efficient acyl donor than gelatin for mTG. Once Z–Gln–Gly was fully utilized, the gelatin could be used as an acyl donor for crosslinking reactions with the remaining -NH_2_, and the Gel–TG–ZQG hydrogel could be formed (as shown in Figure 2i). However, when the ratio of Z–Gln–Gly to the primary amino groups in the gelatin was greater than 0.5, the MSD curves (Figure 2e–g) remained approximately linear throughout the test, suggesting that the tracer particles moved freely at 50 °C because no network structure was formed. This phenomenon arose mainly because Z–Gln–Gly occupied too many acyl acceptors (−NH_2_), preventing mTG-mediated bioconjugation between gelatin chains (as shown in Figure 2j).

### 3.2. Characterization of Gel–TG–ZQG

The TGA and DTG curves of the gelatin-based hydrogels are shown in Figure 3, whereas the corresponding thermal analytical data are presented in Table 2. All the hydrogels exhibited similar thermal behaviors according to their TGA and DTG curves. All the samples exhibited three distinct stages of weight loss over the temperature range studied. The first stage of weight loss, accounting for 10%, occurred between 45 and 150 °C. This finding corresponded to the release of bound water in the hydrogel due to the strong affinity of gelatin for water molecules [32]. The second stage of weight loss occurred between 200 and 450 °C, which was attributed to the decomposition of the hydrogel matrix. This process included the breakdown of weak chemical interactions, such as hydrogen bonding and π–π stacking, and the breakage of the gelatin backbone and isopeptide bond crosslinking by mTG [33]. A comparison of the curves in Figure 3 and the thermal analytical data in Table 2 reveals that some delicate differences remained. The T_d_ and T_m_ values of the hydrogel samples first increased and then decreased with increasing Z–Gln–Gly ratio, and Gel–TG–ZQG (0.25) had the highest T_d_ (189.66 °C) and T_m_ (323.65 °C) values among the samples. The reason for this phenomenon was that the π–π stacking provided by the carbobenzoxy group in Z–Gln–Gly and the covalent crosslinking between the gelatin chains improved the thermal stability [34]. Conversely, excessive amounts of Z–Gln–Gly led to the formation of few covalent bonds, affecting the thermal stability of the hydrogel.

After lyophilization, the hydrogel samples were quenched in liquid nitrogen, and cross sections of the hydrogels were observed by scanning electron microscopy. The results are shown in Figure 4. The microstructure of each hydrogel sample featured a dense and regular honeycomb pore structure. The internal structure of the pore body was smooth, and the pore size distribution was relatively uniform, indicating that the hydrogels had stable network structures. The presence of a continuous network and macroporous structure facilitated the electrical conductivity of hydrogels [35,36]. For Gel–TG–ZQG (0.05), Gel–TG–ZQG (0.125), and Gel–TG–ZQG (0.25), the network structures became increasingly uniform and dense with increasing Z–Gln–Gly, and the mean pore size decreased from 32.64 μm to 22.45 μm. We believed that the microstructure developed as a result of improved π–π stacking and covalent crosslinking, which exerted pressure on the hydrogel matrix, thus facilitating the construction of relatively dense network structures [37]. The mean pore sizes of the hydrogels increased gradually when the ratio of Z–Gln–Gly was greater than 0.5, as shown in Figure 4e–g. This phenomenon occurred because of the lack of crosslinking bonds between the gelatin chains, as discussed previously.

### 3.3. Rheological Properties of Hydrogels

Rheological measurements are usually utilized to confirm the gel state and to characterize the viscoelastic properties of hydrogel systems. In a typical frequency sweep (Figure 5a), the variation in the storage modulus (G′) was measured as a function of frequency under a constant strain at 15 °C. Evidently, the introduction of Z–Gln–Gly significantly increased G′ compared to that resulting from crosslinking by mTG alone, and Gel–TG–ZQG (0.25) had the highest G′. A possible explanation for this result was that with the grafting of Z–Gln–Gly to gelatin, there was a possibility of strong π–π stacking of the carbobenzoxy groups between the gelatin chains, resulting in an increase in the elastic modulus of the polymer [38]. Furthermore, the storage moduli of the gelatin-based hydrogel samples exhibited different frequency dependences. In particular, the G′ value frequency dependence was increasingly pronounced when the ratio of Z–Gln–Gly added exceeded 0.5. As explained by Sarbon et al. [39], an entangled network showed frequency dependence, while a covalent gel showed frequency independence. Therefore, Gel–TG–ZQG (0.5), Gel–TG–ZQG (0.75), and Gel–TG–ZQG (1) were probably formed by π–π stacking. However, among the Gel−TG samples, Gel–TG–ZQG (0.05), Gel–TG–ZQG (0.125), and Gel–TG–ZQG (0.25) exhibited stable covalent crosslinks. This conclusion was consistent with the results of the MS–DWS, TGA, and SEM analyses. Consistent results could also be observed in the G′ temperature sweep curves (Figure 5b). Gel–TG–ZQG hydrogels displayed greater G′ values than Gel−TG hydrogels at low temperatures. In addition, the highest storage modulus was observed for Gel–TG–ZQG (0.25) at temperatures less than 10 °C, at approximately 7800 Pa; this modulus was 2.8 times greater than that of Gel−TG. Importantly, although there was a significant decrease in the G′ value of the hydrogel above 20 °C, Gel–TG–ZQG (0.25) was in the gel state at 5–40 °C, indicating good dimensional stability. Notably, the G′ curve of the Gel–TG–ZQG hydrogels decreased sharply with increasing temperature due to the destruction of π–π stacking [38]. Especially for Gel–TG–ZQG (0.5), Gel–TG–ZQG (0.75), and Gel–TG–ZQG (1), the G′ curve decreased to near 0 Pa, indicating that the network of the gels generated by π–π stacking collapsed and gradually converted into a sol state.

### 3.4. Mechanical Properties of the Hydrogels

Figure 6a shows the compressive stress-strain curves of hydrogels with different Z–Gln–Gly contents. The Gel–TG–ZQG hydrogels exhibited better compressive properties than did the Gel−TG hydrogels when the ratio of Z–Gln–Gly to the primary amino groups in the gelatin was less than 0.25. The maximum compressive strength reached 0.15 MPa at 80% strain, which was obtained from Gel–TG–ZQG (0.25). This finding demonstrated that both the elasticity and strength could be enhanced by introducing Z–Gln–Gly. The main reason for the improved toughness was the presence of π–π stacking interactions, which were incorporated into covalent networks and could efficiently dissipate energy through the rupture and recombination of noncovalent bonds, preventing cracks from expanding during compression [40]. Similar results were reported for other composite hydrogels reinforced with graphene [15], tannic acid [22] and polydopamine [41]. However, an excessive Z–Gln–Gly content led to a lack of covalent crosslinking and resulted in a significant decrease in compressive strength. Similar trends could also be observed in the tensile strength and elongation at break results (Figure 6c). As expected, Gel–TG–ZQG (0.25) exhibited the maximum tensile strength (0.343 MPa) and elongation at break (218.30%). Overall, Gel–TG–ZQG (0.25) exhibited a high mechanical strength and toughness. These results supported our new design strategy in which introducing Z–Gln–Gly into gelatin hydrogels could contribute to energy dissipation, thus improving the mechanical strength and fatigue resistance.

### 3.5. Swelling Behaviors of the Hydrogels

Swelling behavior is a vital characteristic of hydrogels, and different applications of hydrogels have different swelling requirements [42]. The swelling ratios of the hydrogels were measured by the gravimetric method, and the results are shown in Figure 7. Gel–TG–ZQG (0.25) exhibited the lowest swelling ratio among all the samples, approximately 427%. This result arose because the construction of π–π stacking and the presence of covalent crosslinking made the internal structure of the hydrogel increasingly compact, which could be observed via SEM (Figure 4). Figure 7 shows that excessive Z–Gln–Gly addition caused a sharp increase in the swelling ratio of the hydrogels, and for Gel–TG–ZQG (1), the swelling ratio approached 1200%. This behavior could likely be attributed to the weak restriction of the hydrogel polymers as a result of the lack of covalent crosslinking.

### 3.6. Cytotoxicity Levels of the Hydrogels

In vitro cell viability was evaluated by the MTT method. After 24 h and 48 h of culture, cell viability was measured as shown in Figure 8. The results revealed that the percentage of viable cells exceeded 80%, and the relative proliferation rate at 48 h was greater than that at 24 h. These findings indicated that the Gel–TG–ZQG hydrogels had no obvious cytotoxicity. This difference was attributed mainly to the biocompatibility of gelatin and Z–Gln–Gly and the safety of enzymatic crosslinking, which we previously described [25,43]. Since the Gel–TG–ZQG hydrogel showed no obvious cytotoxicity, it was directly contacted with human tissue when fabricated into strain sensors.

### 3.7. Conductivities of the Hydrogels

In electronic devices, hydrogels are usually required to have electrical conductivity. The conductivity was determined by the quantity of conducting ions and the efficiency of ion transport pathways [36]. Herein, the electrical conductivities of hydrogels with different Z–Gln–Gly contents were measured at room temperature, as shown in Figure 9. Adding Z–Gln–Gly and gradually varying the ratio of Z–Gln–Gly to gelatin primary amines from 0.05 to 1 resulted in a gradual increase in conductivity from 7.09 × 10^−5^ S/cm to 12.92 × 10^−3^ S/cm, indicating the enhancing effect of Z–Gln–Gly on hydrogel conductivity. This result occurred because the π–π stacking created by the carbobenzoxy group on Z–Gln–Gly could provide an effective electronic pathway, resulting in good conductivity [23,44]. Conversely, the newly introduced carboxyl groups of Z–Gln–Gly could act as ion donors to improve hydrogel conductivity.

### 3.8. Strain Sensitivities of Hydrogel Strain Sensors

Owing to its excellent mechanical properties, sufficient dimensional stability and electrical conductivity, Gel–TG–ZQG (0.25) was fabricated as a flexible strain sensor that could detect various human motions. Importantly, high sensitivity and rapid response are important characteristics of a strain sensor. Owing to the improved toughness and effective electronic pathway caused by π–π stacking, the Gel–TG–ZQG (0.25) strain sensor exhibited a rapid response and recovery time and high sensitivity. As shown in Figure 10a, the response time to pressure loading was 260.37 ms, and the recovery time was 130.02 ms. Compared with related works (Table 3), the Gel–TG–ZQG (0.25) strain sensor prepared in this work achieved good results. The sensitivities were 0.138 and 0.093 kPa^−1^ when the applied pressure was within the ranges of 0–2.3 and 2.3–5 kPa, respectively (Figure 10b). As with most reported hydrogel pressure sensors, the pressure sensitivity of the sensor was composed of more than one region [45,46]. This result arose because when the pressure reached a certain level, the deformation caused by the molecular chain in the hydrogel in the original crimped state was limited, resulting in a decrease in the pressure sensitivity [47].

To verify the sustained stability and strain sensitivity of the Gel–TG–ZQG (0.25) strain sensor, the rate of change in the resistance under different pressure strains was evaluated via cyclic loading. The ΔR/R_0_ signals of the hydrogel sensor under various pressure strains are shown in Figure 10c. To ensure long-term use, the durability and stability of the sensor were essential. The ΔR/R_0_ signals exhibited periodic cyclic changes, and there was no obvious delaying phenomenon during the loading/unloading process. This result occurred because the porous network structure and mechanical toughness of Gel–TG–ZQG (0.25) enabled effective stress alleviation during long-term cycling. At strains of 1, 2 and 5 kPa, the relative resistance changes were approximately 3%, 15% and 65%, respectively. The resistance change rate increased with increasing pressure strain, indicating high strain sensitivity. Moreover, the strain sensor could still accurately respond to low-pressure loading and unloading through resistance changes, which was beneficial for monitoring subtle human activities.

To demonstrate the practical application of Gel–TG–ZQG (0.25) as a strain sensor, it was directly attached to various parts of the human body to monitor human activities in real time. As shown in Figure 10d, the strain sensor was conformally attached to the throat to detect the motion of repeated swallowing. The variations in the real-time resistance of the sensor were consistent with the changes in pressure caused by swallowing, and the electrical signals showed high stability and repeatability. In addition, even subtle strains from the movements of the epidermis and muscles of the neck could be detected. The characteristic peak shape produced by the electrical signals allowed us to distinguish the type of movement. Similarly, the bending force of the complete tension–relaxation cycles during fist clenching (Figure 10e) and the bending–releasing motion of the knee during walking (Figure 10f) could be accurately detected. The corresponding curves showed that the current values were almost uniform during the same motion. Figure 10g and h show that the sensor exhibited significantly different responses to gentle and brutal finger presses, indicating that the strain sensor could effectively monitor repeated finger pressing with stable response signals. However, irrespective of the detection attempt, the relative resistances of the sensors were able to fully recover to the pristine state when human activity ceased. Therefore, the Gel–TG–ZQG (0.25) sensor showed high potential as an on-skin electronic for collecting bioelectricity signals and for use in human–computer interfaces.

## 4. Conclusions

In summary, gelatin-based hydrogels with excellent mechanical performance and conductivity were fabricated by integrating Z–Gln–Gly into gelatin hydrogels via enzymatic crosslinking. The dynamic π–π stacking and covalent crosslinking afforded Gel–TG–ZQG (0.25) excellent mechanical properties, including a high storage modulus, robust elasticity and improved fracture toughness. In addition, due to the occurrence of π–π stacking, the gelatin-based hydrogel exhibited suitable conductivity and excellent sensing performance. The strain sensor based on Gel–TG–ZQG (0.25) showed a rapid response/recovery time and high sensitivity. Furthermore, the hydrogel had no obvious cytotoxicity because of the biocompatibility of gelatin or Z–Gln–Gly and because of the safety of enzymatic crosslinking, which ensured that it could be in direct contact with human tissue. The Gel–TG–ZQG (0.25) hydrogel was developed as a strain sensor for detecting various human motions, and it performed well. Therefore, the gelatin-based conductive hydrogel prepared in this study had good application potential in biosensors and wearable electronic devices.

## Figures and Tables

**Figure 1 polymers-16-00999-f001:**
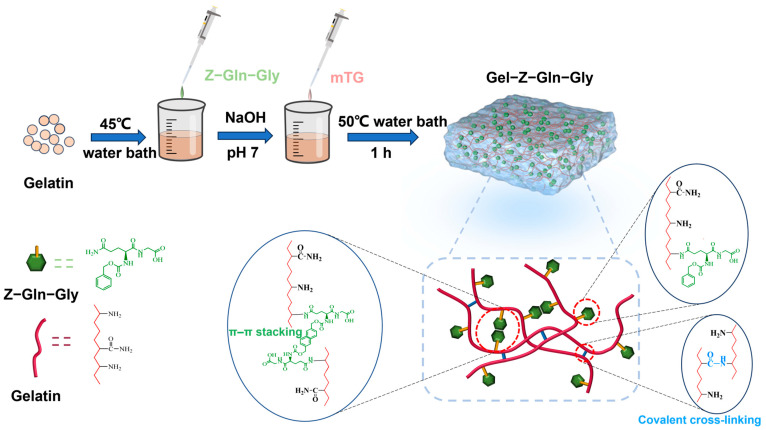
Schematic of the formation of the Gel–TG–ZQG hydrogel and the interactions between the side groups in the hydrogel.

**Figure 2 polymers-16-00999-f002:**
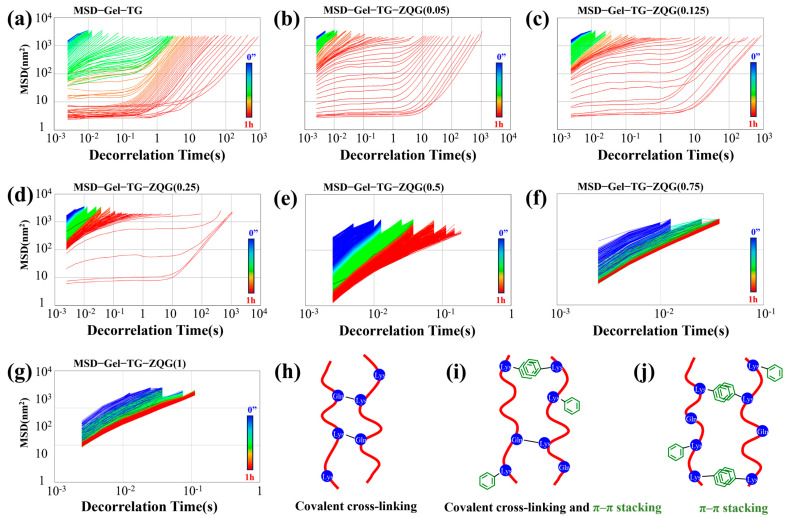
MSD curves collected at 50 °C during Gel–TG–ZQG hydrogel formation. (**a**) Gel−TG, (**b**) Gel–TG–ZQG (0.05), (**c**) Gel–TG–ZQG (0.125), (**d**) Gel–TG–ZQG (0.25), (**e**) Gel–TG–ZQG (0.5), (**f**) Gel–TG–ZQG (0.75) and (**g**) Gel–TG–ZQG (1). (**h**) Schematic diagram of the covalent crosslinks between gelatin chains (for Gel−TG). (**i**) Schematic diagram of the interactions between gelatin chains including covalent crosslinking and π–π stacking (for Gel–TG–ZQG (0.05), Gel–TG–ZQG (0.125), and Gel–TG–ZQG (0.25)). (**j**) Schematic diagram of the π–π stacking interactions between gelatin chains (for Gel–TG–ZQG (0.5), Gel–TG–ZQG (0.75), and Gel–TG–ZQG (1)).

**Figure 3 polymers-16-00999-f003:**
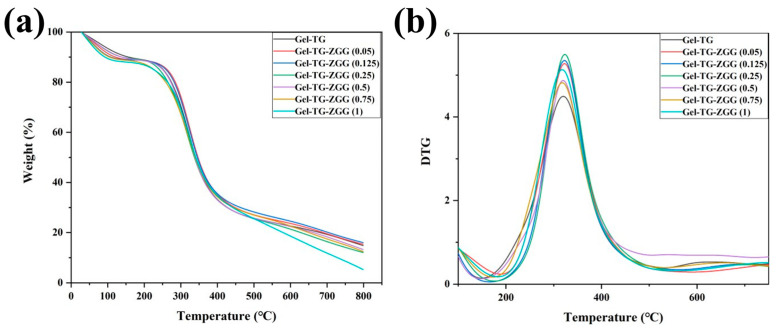
TGA (**a**) and DTG (**b**) curves of the Gel−TG and Gel–TG–ZQG hydrogels.

**Figure 4 polymers-16-00999-f004:**
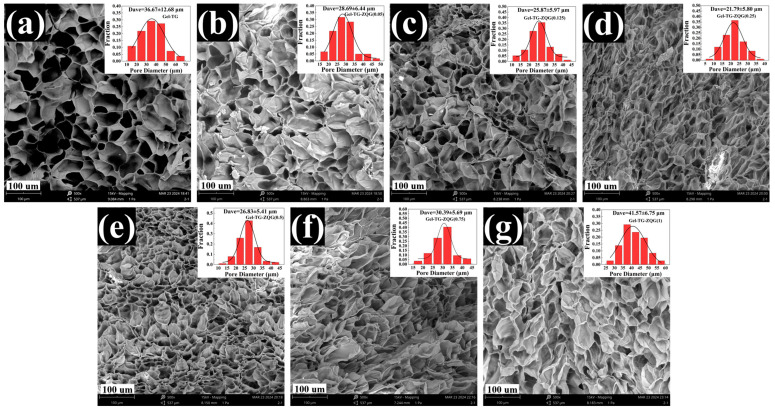
Microstructures and pore size distributions of the freeze-dried hydrogels. (**a**) Gel−TG, (**b**) Gel–TG–ZQG (0.05), (**c**) Gel–TG–ZQG (0.125), (**d**) Gel–TG–ZQG (0.25), (**e**) Gel–TG–ZQG (0.5), (**f**) Gel–TG–ZQG (0.75), and (**g**) Gel–TG–ZQG (1).

**Figure 5 polymers-16-00999-f005:**
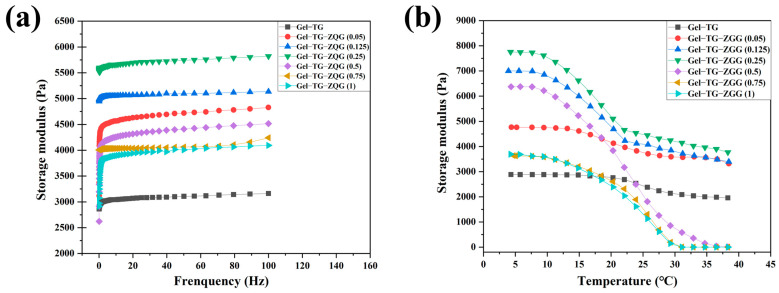
Storage moduli of the gelatin-based hydrogels tested under a frequency sweep at 15 °C (**a**) and under a temperature sweep (**b**).

**Figure 6 polymers-16-00999-f006:**
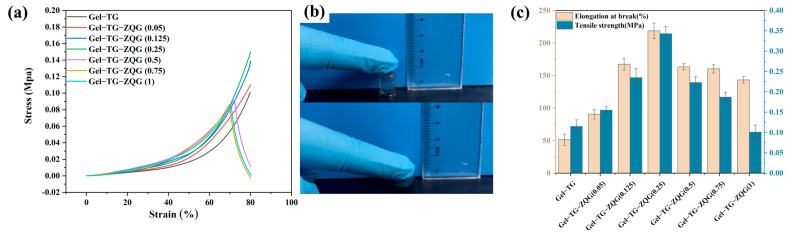
Mechanical properties of the gelatin-based hydrogels. (**a**) Compressive stress-strain curves of hydrogels with different contents of Z–Gln–Gly. (**b**) Typical images of Gel–TG–ZQG (0.25) before and after compression. (**c**) Comparison of the tensile strength and elongation at break values of hydrogels with different Z–Gln–Gly contents.

**Figure 7 polymers-16-00999-f007:**
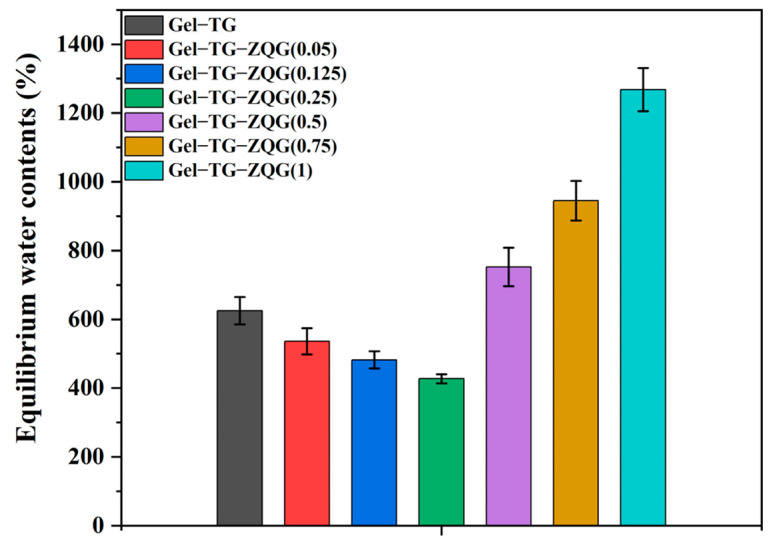
Equilibrium swelling ratios of the Gel−TG and Gel–TG–ZQG hydrogels in PBS solution at pH 7.4.

**Figure 8 polymers-16-00999-f008:**
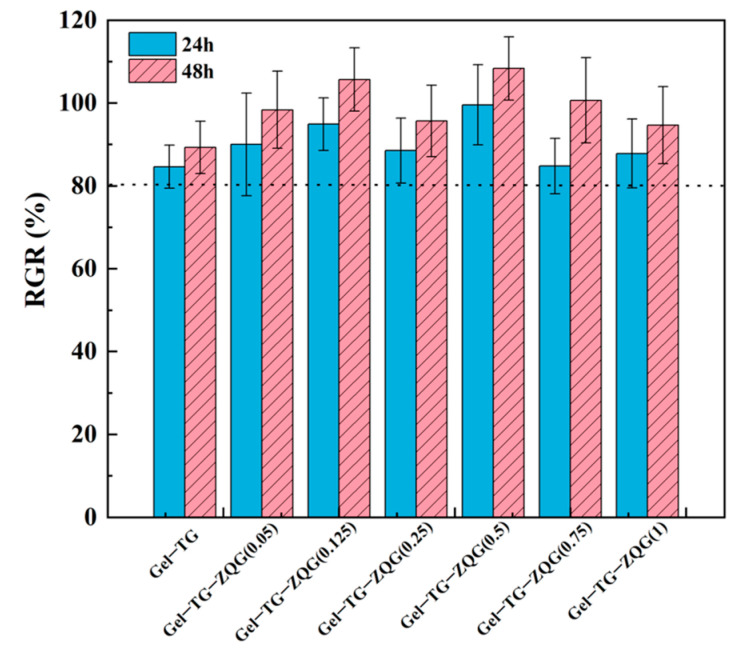
Viability of HeLa cells cultured on media supplemented with different hydrogel extracts for 24 h and 48 h.

**Figure 9 polymers-16-00999-f009:**
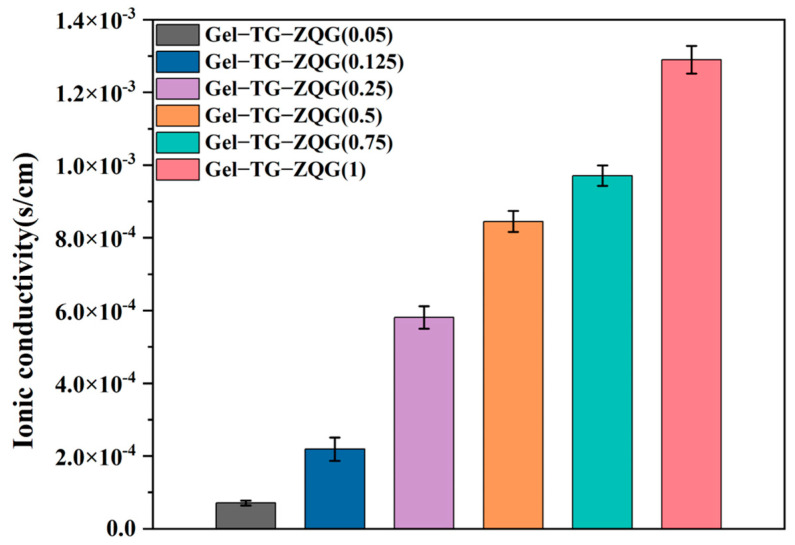
Electrical conductivities of the Gel–TG–ZQG hydrogels.

**Figure 10 polymers-16-00999-f010:**
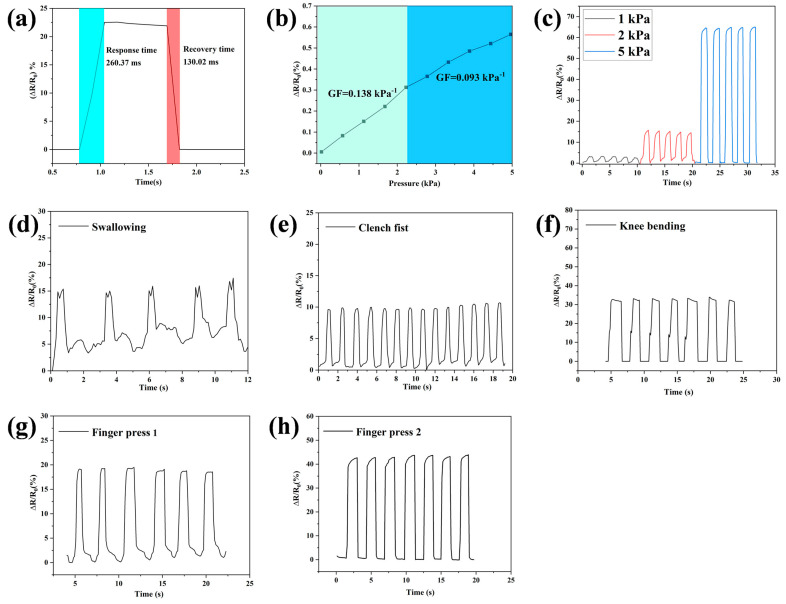
(**a**) Response/recovery time of the Gel–TG–ZQG (0.25) pressure sensor. (**b**) Sensitivity of the Gel–TG–ZQG (0.25) pressure sensor. (**c**) Curves of relative resistance versus time under different strains. Relative resistance versus time based on Gel–TG–ZQG (0.25) pressure sensors for cyclic motion at the throat (**d**), fist €, knee joints (**f**), and fingers (**g**,**h**).

**Table 1 polymers-16-00999-t001:** Ratio relationship between Z–Gln–Gly (ZQG) and gelatin in the hydrogel samples.

Samples	Gelatin		ZQG		Molar Ratio of ZQG to the Primary Amines of Gelatin
	Mass (g)	Primary Amines (μmol)	Mass (mg)	Mole (μmol)	
Gel−TG	0.79	498.03	0	0	0
Gel–TG–ZQG (0.05)	0.79	498.03	8.40	24.90	0.05
Gel–TG–ZQG (0.125)	0.79	498.03	21.00	62.25	0.125
Gel–TG–ZQG (0.25)	0.79	498.03	42.00	124.51	0.25
Gel–TG–ZQG (0.5)	0.79	498.03	84.00	249.01	0.5
Gel–TG–ZQG (0.75)	0.79	498.03	126.00	373.52	0.75
Gel–TG–ZQG (1)	0.79	498.03	168.00	498.03	1.00

**Table 2 polymers-16-00999-t002:** Temperature of 50% weight loss (T_50_), temperature of initial degradation (T_d_), and temperature corresponding to the maximum rate of weight loss (T_m_) of each hydrogel.

Samples	T_50_ (°C)	T_d_ (°C)	T_m_ (°C)
Gel−TG	341.65	173.06	318.02
Gel–TG–ZQG (0.05)	341.08	175.75	319.98
Gel–TG–ZQG (0.125)	345.09	183.14	322.98
Gel–TG–ZQG (0.25)	347.06	189.66	323.65
Gel–TG–ZQG (0.5)	338.58	166.54	322.32
Gel–TG–ZQG (0.75)	339.55	149.31	319.75
Gel–TG–ZQG (1)	337.18	148.95	317.67

**Table 3 polymers-16-00999-t003:** Comparison of the response and recovery times of hydrogel sensors reported in the literature.

Samples	Response Time	Recovery Time	References
G-PPy/PAAm-TA	400 ms	400 ms	[22]
HK/PVA/NaCl	170 ms	191 ms	[45]
PAAm/Gelatin	200 ms	600 ms	[47]
PAAm	150 ms	400 ms	[48]
PAM/SA	800 ms	/	[49]
PVA/PVP/Ti3AlC2	233 ms	233 ms	[50]
AAm/HMAm/β-CD	210 ms	130 ms	[51]
Gel–TG–ZQG (0.25)	260.37 ms	130.02 ms	This work

## Data Availability

The data presented in this study are available on request from the authors.

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
