# Peer review of "Mechanically Tough and Conductive Hydrogels Based on Gelatin and Z–Gln–Gly Generated by Microbial Transglutaminase"

_polymers, 2024, doi:10.3390/polym16070999_

Round 1
Reviewer 1 Report
Comments and Suggestions for Authors
The presented paper is very interesting and develops a new class of materials synthesized using enzymes. On the other side, the fabrication of the conductive and biocompatible polymer systems is not trivial and hard work. The paper can be accepted after major revision; some issues should be clarified.
The authors concentrate their attention on synthesized hydrogels as biosensors; however, these systems have much stronger potential for application, which should be highlighted in the introduction. I suggest using and citing the review: https://doi.org/10.1002/tcr.202300217, where the potential of conductive systems for biomedical applications was summarized.
The preparation of gelatin-based hydrogels should be explained in detail. It is completely unclear how carbobenzoxy-l-glutamine glycine molecules were attached to gelatin macromolecules, was it catalyzed by transglutaminase? The nomenclature Gel-TG-ZQG (0.05) and others are hard to understand. Please add a table that exactly shows what the ratio is between Z-Gln-Gly and gelatine. The authors noted, "According to the molar ratio of Z-Gln-Gly to the primary amines of gelatin..."; however, it is unclear what concentration of the primary amines is in the gelatine. It is not clear how much croslincking was formed and carbobenzoxy-l-glutamine glycine molecules attached to the gelatin hydrogel.
The authors used the multispeckle diffusing wave spectroscopy (MS–DWS) technique for the characterization of hydrogel formation. This is a new and interesting methodology, but additionally, it will be valuable to add traditional FTIR spectroscopy.
By looking at the gel's properties, a strong link was found between the number of Z-Gln-Gly molecules and the formation of cross-links in hydrogels. Illustrating this relationship in a figure would enhance the understanding of the readers.
The authors noticed an interesting phenomenon: "...G' curve of the Gel-TG-ZQG hydrogels decreased sharply with increasing temperature due to the destruction of π–π stacking...". The discussion on this topic should be significantly expanded, because this finding opens up significant opportunities in the development of a new class of sensitive hydrogels.
Swelling behaviour is strange and should be explained in a more appropriate manner. If this behavior could likely be attributed to the weak restriction of the hydrogel polymers as a result of the lack of covalent crosslinking, why is the water content decreasing to some value?
The quality of Figures 6 and 10 should be improved.
Comments on the Quality of English LanguageMinor editing of English language required.
Reviewer 2 Report
Comments and Suggestions for Authors
1. The manuscript is too complex to understand due to the numerous parameters related to hydrogel production, making it difficult to comprehend the context.
2. How does the hydrogel be assembled into a strain sensor? More information should be provided.
3. It is confusing that the remaining weight of a natural hydrogel for TGA analysis over 800 ºC is around 20%.
4. According to Figure 5b, Gel-TG remains in a gel state at different temperatures. Even though the π–π stacking collapsed, the TG-induced crosslinking of gelatin remains. Why did the G’ decrease to near 0 Pa for Gel-TG-ZQG(0.5), Gel-TG-ZQG(0.75), and Gel-TG-ZQG(1) at high temperatures?
5. Why were the cell viability tests performed on the cancer cells? It is also noteworthy that without ZQG, the cell viability of hydrogel is only 85%. It seems like the hydrogel has cytotoxicity.
6. For human motion sensing, it’s better to have a visual picture showing how the hydrogel works.
7. As a comparison, Table 2 should include the data of Gel-TG-ZQG (0.25).
8. Figures 2,4,6,10 are very vague and affect the readability of the text.
9. Ensure that abbreviations are defined upon their first mention and maintain consistency throughout.
Comments on the Quality of English LanguageMinor revision by a native English-speaking scientist is mandatory.
Round 2
Reviewer 1 Report
Comments and Suggestions for Authors
The authors completely answered all questions. The paper can be accepted in its present form.
Comments on the Quality of English LanguageMinor editing of English language required.
